# Using Dual Toll-like Receptor Agonism to Drive Th1-Biased Response in a Squalene- and α-Tocopherol-Containing Emulsion for a More Effective SARS-CoV-2 Vaccine

**DOI:** 10.3390/pharmaceutics14071455

**Published:** 2022-07-12

**Authors:** Kristopher K. Short, Stephanie K. Lathrop, Clara J. Davison, Haley A. Partlow, Johnathan A. Kaiser, Rebekah D. Tee, Elizabeth B. Lorentz, Jay T. Evans, David J. Burkhart

**Affiliations:** 1Center for Translational Medicine, University of Montana, Missoula, MT 59812, USA; kristopher.short@umconnect.umt.edu (K.K.S.); stephanie.lathrop@mso.umt.edu (S.K.L.); clara.davison@mso.umt.edu (C.J.D.); haley.partlow@mso.umt.edu (H.A.P.); johnathan.kaiser@mso.umt.edu (J.A.K.); rebekah.tee@mso.umt.edu (R.D.T.); elizabeth.lorentz@mso.umt.edu (E.B.L.); jay.evans@mso.umt.edu (J.T.E.); 2Department of Biomedical and Pharmaceutical Sciences, University of Montana, Missoula, MT 59812, USA

**Keywords:** COVID-19, SARS-CoV-2, subunit vaccine, adjuvant, toll-like receptor (TLR) agonist, TLR4, TLR7/8, AS03, MF59

## Abstract

A diversity of vaccines is necessary to reduce the mortality and morbidity of SARS-CoV-2. Vaccines must be efficacious, easy to manufacture, and stable within the existing cold chain to improve their availability around the world. Recombinant protein subunit vaccines adjuvanted with squalene-based emulsions such as AS03™ and MF59™ have a long and robust history of safe, efficacious use with straightforward production and distribution. Here, subunit vaccines were made with squalene-based emulsions containing novel, synthetic toll-like receptor (TLR) agonists, INI-2002 (TLR4 agonist) and INI-4001 (TLR7/8 agonist), using the recombinant receptor-binding domain (RBD) of SARS-CoV-2 S protein as an antigen. The addition of the TLR4 and TLR7/8 agonists, alone or in combination, maintained the formulation characteristics of squalene-based emulsions, including a sterile filterable droplet size (<220 nm), high homogeneity, and colloidal stability after months of storage at 4, 25, and 40 °C. Furthermore, the addition of the TLR agonists skewed the immune response from Th2 towards Th1 in immunized C57BL/6 mice, resulting in an increased production of IgG2c antibodies and a lower antigen-specific production of IL-5 with a higher production of IFNγ by lymphocytes. As such, incorporating TLR4 and TLR7/8 agonists into emulsions leveraged the desirable formulation and stability characteristics of emulsions and can induce Th1-type humoral and cell-mediated immune responses to combat the continued threat of SARS-CoV-2.

## 1. Introduction

The emergence of the virus SARS-CoV-2, and its resulting respiratory illness COVID-19, has led to unprecedented increases in morbidity and mortality [1] and economy instability [2]. To combat the SARS-CoV-2 pandemic and COVID-19, two mRNA-based vaccines, BioNTech-Pfizer’s BNT162b2 and Moderna’s mRNA-1273, marched with an unprecedented pace from an IND designation through phase I/II/III trials to a US FDA emergency use authorization (EUA) in December 2020 [3,4], over an incredible 11 months after the identification of SARS-CoV-2 and the publication of its genetic sequence [5]. These vaccines cause adaptive immunity to the in situ production of the SARS-CoV-2 S (spike) protein, which has resulted in greater than 90% prevention of COVID-19 after SARS-CoV-2 infection in phase III trials [6,7]. These vaccines target the spike protein, which binds to angiotensin-converting enzyme 2 (ACE2) on the target cells via the spike protein’s receptor-binding domain (RBD) [8] and mediates viral fusion with the host cell [9].

Despite the early success of these vaccines, there is still a critical unmet need for other vaccine approaches. Recently, the emergence of highly transmissible genetic variants of SARS-CoV-2 containing mutations in the spike protein have resulted in a lower efficacy of the currently approved vaccines [10,11] and increased numbers of breakthrough infections [12]. Additionally, the waning immunity of the current approaches reiterates the need for new, highly effective vaccines that would ideally offer an improved resistance to variants and breakthrough infections [13] as well as a vastly improved stability for deployment across the globe.

Newer vaccines, including adjuvanted recombinant protein subunit vaccines, are currently in late-stage clinical trials across the globe [14]. These vaccines pair an adjuvant that improves the type and quality of the immunological response with a recombinant protein antigen. Squalene-based oil-in-water (O/W) emulsions are adjuvants with demonstrated success in the clinic [15,16] and are already in late-stage clinical trials for use in SARS-CoV-2 vaccines [17,18]. Previously, they have been studied extensively in the context of influenza vaccines, where they increased the antibody titer and affinity [19], enhanced cross protection against variants [20,21], and reduced the effective antigen dose [22], which were crucial to the response during the 2009 H1N1 influenza pandemic. Additionally, squalene-based O/W emulsions are already produced at a low cost and are stable for months to years, which enhances their deployment to developing countries [23].

The squalene-based O/W emulsion AS03™ has been shown to generate high antigen-specific antibody titers against the SARS-CoV-2 spike protein and RBD antigens in experimental subunit vaccines [24,25]. While high levels of neutralizing antibodies are effective in preventing infection by SARS-CoV-2, data increasingly point to the importance of the CD4^+^ T cell response in the protection against disease [26,27,28]. Studies of individuals who recovered from the SARS-CoV-1 outbreak in 2003 have revealed that memory T cells against the spike and nucleocapsid proteins are still detectable 17 years after the infection, giving reason to believe that a memory T cell response is key in the development of long-lasting protection [29]. Additionally, T cell epitopes are often found in more conserved regions of proteins, and therefore T cell protection may extend across viral variants. Cross-reactivity to SARS-CoV-2 was observed with some memory T cells from individuals that had recovered from severe acute respiratory syndrome (SARS) [29], and a higher number of T cells that were reactive against SARS-CoV-2 proteins was shown to correlate with a lessened development of the disease in previously uninfected patients with a known exposure to SARS-CoV-2 [30,31].

While AS03™ and other squalene-based O/W emulsions enhance T cell development along with enhancing antibody titers, these tend to be mixed or preferentially skewed towards Th2 responses [25,32]. However, effective respiratory viral clearance relies on the development of a Th1-mediated and accompanying cytotoxic T cell response [33,34]. Squalene-based O/W emulsions are not only effective adjuvants, but are also versatile and effective delivery vehicles for additional adjuvants such as naturally-derived or synthetic toll-like receptor (TLR) 4 agonists and lipidated TLR7/8 agonists [35]. Since TLR4 and TLR7/8 agonists are known to promote cell-mediated immune responses alone or in combination [35,36], incorporating TLR agonists into squalene-based emulsions has been demonstrated to improve both the humoral and cell-mediated responses compared to the emulsion alone [37]. While most previous research has focused on single TLR4 or TLR7/8 agonist incorporation in squalene-based emulsions, we propose that squalene-based emulsions can be used to dually deliver TLR4 and lipidated TLR7/8 agonists. As such, this approach may successfully marry the stability advantages of emulsions with the adjuvant effect of TLR agonism for use in SARS-CoV-2 vaccines.

To test this hypothesis, we sought to incorporate INI-2002, a novel synthetic TLR4 agonist, and INI-4001, a novel lipidated oxoadenine and TLR7/8 agonist, into AddaS03-like emulsions singly and in combination. The resulting emulsions were characterized by their physical properties, including the emulsion’s hydrodynamic diameter, polydispersity (PDI), zeta potential, agonist recovery after filtration, and colloidal stability during storage. The adaptive immune response to these emulsions was characterized in C57BL/6 mice by measuring the antigen-specific antibody titers and T cell cytokine production after immunization. Additionally, the ability of serum antibodies to neutralize variants of SARS-CoV-2 was determined by a surrogate virus neutralization test (sVNT).

## 2. Materials and Methods

### 2.1. Materials

All the materials used to prepare the emulsions were of research grade and used as received. Squalene was purchased from MP Biomedicals (Irvine, CA, USA), DL-α-tocopherol from TCI (Portland, OR, USA), Tween-80 from Spectrum Chemical (New Brunswick, NJ, USA), and Span-85 from Sigma-Aldrich (St. Louis, MO, USA). The TLR agonists were synthesized and processed to over 99% purity as described previously [38,39,40,41]. DPBS, citrate, sterile water for irrigation (WFI), assay buffers, and analytical-grade reagents for HPLC were purchased through Thermo Fisher Scientific (Waltham, MA, USA).

### 2.2. Emulsion Selection and Preparation

An emulsion system containing 2.5% *v*/*v* squalene, 2.5% *v*/*v* DL-α-tocopherol, and 0.96% *v*/*v* Tween-80 in DPBS with a pH of 6.8 was chosen as the emulsion vehicle for this study, due to the success of AS03™ in other vaccines [42] and its continued testing for use as an adjuvant in a SARS-CoV-2 vaccine [17]. The commercially available AddaS03™ is a research-grade equivalent of AS03™ and also thoroughly researched. Since the emulsions used here were made outside of a commercial manufacturing setting with reagent-grade materials, we refer to the emulsions as “AddaS03-like”. Similarly, a positive control emulsion was made from 2.5% *v*/*v* squalene, 0.25% *v*/*v* Tween-80, and 0.25% *v*/*v* Span-85 in a 10 mM sodium citrate buffer with a pH of 6.0, due to the success of MF59™ [43] and its commercially available equivalent AddaVax™ (Invivogen, San Diego, CA, USA). Here, we denote a similar emulsion made from research-grade materials as “MF59-like”.

INI-4001 and INI-2002 were incorporated into emulsions singly or coencapsulated. We targeted concentrations of 0.2 or 2.0 mg/mL of INI-4001 (0.175 or 1.75 mM, respectively) in the final formulation, and INI-2002 was targeted to equimolar amounts (0.175 or 1.75 mM). These concentrations were based on previous results using similar TLR agonists in liposome formulations [36]. The emulsions were made by a modified thin-film hydration method where all the lipids were combined in 9:1 *v*/*v* isopropanol:ethanol. After solvent removal using a Savant SpeedVac vacuum concentrator (Thermo Fisher Scientific), buffer was added under bath sonication at 45–50 °C, and ultra-acoustic focusing with a Covaris S2 Ultrasonicator (Woburn, MA, USA) was used to make a crude emulsion. The Covaris S2 settings were 7 s of processing at 10% duty cycle, an intensity of 10 (~40–50 W), and 1000 cycles per burst followed by 8 s of rest for a total of 12 cycles. The resulting crude emulsion was processed by high-pressure homogenization using six passes at 25,000 PSI with a Microfluidics LV1 (Westwood, MA, USA), which is similar to the processing conditions of other clinically relevant emulsions [44,45,46]. The emulsions were terminally sterile-filtered using a 13 mm Millex GV PVDF filter with a pore size of 0.22 μm (MilliporeSigma, Burlington, MA, USA).

To maintain sterility and to avoid unwanted TLR4 agonism by endotoxins, all emulsions were prepared in a BioChemGARD Biosafety Cabinet (The Baker Company, Sanford, ME, USA) with aseptic technique, endotoxin-free consumables, and depyrogenated glassware, which was heated to 250 °C for 60 min in a BlueM Lab Oven (New Columbia, PA, USA). The LV1 was chemically depyrogenated by soaking in a solution of 0.4% sodium hydroxide in 95% *v*/*v* ethanol for 30 min, followed by rinsing with WFI and then equilibrated with buffer.

### 2.3. Qualitative Characterization of Emulsions

The electron microscopy was performed at the Multiscale Microscopy Core with technical support from the Oregon Health & Science University (OHSU)/FEI Living Lab and the OHSU Center for Spatial Systems Biomedicine. The samples were imaged using a FEI Talos Arctica system with a FEI Ceta 16M CMOS camera (both from Thermo Fisher Scientific). The samples were prepared by transferring them to Quantifoil EM grids and freezing them in liquid ethane using a Vitrobot prior to imaging. When necessary, the samples were diluted with the sample buffer. The images were processed using the Fiji distribution of ImageJ (NIH, Bethesda, MD, USA) [47].

### 2.4. Physical Characterization of Emulsions

Dynamic light scattering (DLS) was used to measure the hydrodynamic diameter, PDI, and zeta potential with a Zetasizer Nano-ZS (Malvern Panalytical, Malvern, UK). For the hydrodynamic diameter and PDI, each sample was measured at stable conditions after a 1:20 dilution in DPBS with a pH of 6.8, and three measurements per sample were averaged for four replicate samples. The hydrodynamic diameter is given based on the intensity function (Z.avg). The zeta potential was measured after a 1:40 sample dilution in 10 mM NaCl at 25 °C. The formulation’s pH was measured from the samples diluted for the zeta potential measurements using an Accumet AB150 pH meter (Thermo Fisher Scientific) and an InLab Micro probe (Mettler-Toledo, Columbus, OH, USA) after a three-point calibration using pH 4.01, 7.00, and 10.01 standards.

### 2.5. TLR Agonist Recovery after Sterile Filtration

The concentrations of INI-4001 and INI-2002 were determined by RP-HPLC using an Acquity Arc separations module and a 2998 PDA Detector (both from Waters, Milford, MA, USA). The emulsions and compound standards were dissolved in 9:1 *v*/*v* tetrahydrofuran:methanol, injected on an XSelect CSH C18 3.5 μm 2.1 × 150 mm column with an XSelect CSH C18 3.5 μm 2.1 × 5 mm guard column (both from Waters), and separated by gradient elution at 0.8 mL/min at 70 °C. Aqueous mobile phase A (MPA) consisted of 60% *v*/*v* acetonitrile and 40% *v*/*v* water buffered with 10 mM ammonium formate that was adjusted to a pH of 3.2 using concentrated formic acid. Organic mobile phase B (MPB) consisted of 50% *v*/*v* acetonitrile, 49% *v*/*v* isopropanol, and 1% *v*/*v* ammonium formate (10 mM final concentration, with a pH of 3.2). The following reverse phase gradient, given in T(min), was run for each sample injection: T0.0 %A: 85, %B: 15; T0.5 %A: 85, %B: 15; T5.0 %B: 100; T6.0 %B: 100; T6.1 %A: 85, %B: 15; and T8.0 %A: 85, %B: 15. The INI-2002 and INI-4001 absorbances were measured at 210 nm and 254 nm, respectively. The compound concentrations were quantified by the peak area interpolated from a seven-point dilution series of the corresponding standard. The percent recovery of INI-2002 and INI-4001 was determined by comparing the concentrations of the samples pre- and post-sterile filtration. MarvinSketch software (ChemAxon, Cambridge, MA, USA) was used to estimate the hydrophilic–lipophilic balance (HLB) values of the TLR agonists using the Griffin [48,49] and Davies [50] estimates.

### 2.6. In Vivo Mouse Model and Immunization

Female 7–12-week-old C57BL/6 mice (The Jackson Laboratory, Bar Harbor, ME, USA) were used for the in vivo studies. The mice were housed in an AAALAC-accredited facility, and all procedures were performed in accordance with the University of Montana’s IACUC-approved animal use protocol, 015-19JEDBS. The mice were immunized twice, fourteen days apart, by intramuscular injections in the hind limb. The injections consisted of the emulsion admixed with 1 μg of recombinant SARS-CoV-2 RBD and brought to 50 μL with the emulsion buffer. The RBD amino acid sequence and production is described elsewhere [51].

The RP-HPLC results were used to inform the dosing of INI-2002, INI-4001, and squalene for each study group (Appendix A). INI-4001 was dosed at 1.0 µg (low dose, L, 0.876 nmol) or 10 µg (high dose, H, 8.76 nmol) per immunization. INI-2002 was dosed at 0.876 nmol or 8.76 nmol per immunization, which was equimolar to the low and high doses of INI-4001, respectively. In coencapsulated emulsions, INI-4001 was maintained at 8.76 nmol (high dose), and INI-2002 was dosed at low (0.876 nmol) or high (8.76 nmol) doses per immunization. These combinations resulted in molar ratios of 1:1 and 10:1 for INI-4001:INI-2002.

The squalene content was measured separately by HPLC using a 2695 separations module and a 2489 UV/Vis detector (both from Waters). The emulsions and compound standards were dissolved in 9:1 *v*/*v* tetrahydrofuran:methanol, injected on a Symmetry C18 3.5 μm 4.6 × 75 mm column with a Phenomenex C18 guard column (both from Waters), and separated using an isocratic method at 0.8 mL/min at 40 °C. The single mobile phase was 91.9% *v*/*v* methanol, 8.0% *v*/*v* isopropanol, and 0.1% *v*/*v* glacial acetic acid, which was modified from elsewhere [52]. The squalene absorbance was measured at 210 nm, and the concentration was quantified by the peak area interpolated from a seven-point dilution series of a standard. The squalene dose was equalized at 144 µg per immunization by adding an AddaS03-like emulsion vehicle if necessary. This squalene dose falls within the range of doses (42.7–213.6 µg) used in previous mouse studies with AS03™ [53], and is equivalent to 1.3% of the 10.68 mg human dose. The MF59-like emulsion control was dose-matched to 144 µg of squalene per immunization.

### 2.7. In Vivo Serum Antibody Titer

The serum was analyzed by ELISA for RBD-specific total IgG, IgG1, and IgG2c antibody titers fourteen days post-secondary immunization. The mice were euthanized followed by a blood collection via cardiac puncture. The serum was separated using Microtainer Serum Separator Tubes (BD Biosciences, San Jose, CA, USA) and stored at −20 °C. The serum was diluted to between 1:50 and 1:1000 in an EIA buffer according to the expected antibody response. Nunc MaxiSorp 96-well plates (Thermo Fisher Scientific) were coated overnight at 25 °C with 100 µL of the RBD antigen at 2.5–5.0 µg/mL, washed three times with 1× PBS + 0.05% Tween-20, and blocked with SuperBlock (ScyTek Laboratories, Logan, UT, USA) for 1 h at 37 °C. The plates were then incubated with diluted serum for 2 h at 37 °C, followed by binding with anti-mouse IgG-, IgG1- or IgG2c-HRP secondary antibodies (Bethyl Laboratories, Montgomery, TX, USA). Detection was performed using TMB Substrate (BD Biosciences). The absorbance was measured at 450 nm using a SpectraMax 190 microplate reader (Molecular Devices, San Jose, CA, USA), and the antibody titers were reported as the dilution factor that would result in an optical density of 0.30. Any measurements below the limit of detection when the sera were diluted to 1:50 were given a nominal value of 1.0 for the analysis.

### 2.8. Surrogate Virus Neutralization Test (sVNT)

Serum collected fourteen days post-secondary immunization was used for the inhibition of soluble ACE2 binding to SARS-CoV-2 RBD as a measure of its virus neutralization ability. Sera from individual mice were assayed for their ability to block the binding of recombinant human ACE2 to a purified RBD or spike protein by a V-PLEX SARS-CoV-2 ACE2 Panel 6 Kit K15436U (Meso Scale Diagnostics, Rockville, MD, USA), and electrochemiluminescence arising from the bound ACE2 protein was detected by the MESO QuickPlex SQ 120 imager (Meso Scale Diagnostics). The percent inhibition was calculated according to the manufacturer’s instructions.

### 2.9. Ex Vivo RBD-Specific Cytokine Output

The draining lymph nodes (DLN, inguinal and popliteal) were collected following the sacrifice at fourteen days post-secondary immunization. The DLN cells were isolated by mechanical disruption, and 200,000 cells from each animal were plated per well in 96-well plates and restimulated with 10 µg/mL of RBD in RPMI containing 10% FBS for 72 h. When necessary, the cells from two animals were pooled together in order to achieve sufficient cell numbers. At 72 h, the cells were centrifuged for 5 min at 400× *g*, and the supernatants were collected and stored at −20 °C until they were assayed for RBD-specific IFNγ, IL-17A, and IL-5 cytokines. The cytokine concentration was determined using a U-Plex multiplex ELISA kit (Meso Scale Diagnostics) according to the manufacturer’s instructions.

### 2.10. Data Analysis and Graphing

GraphPad Prism 9 software (San Diego, CA, USA) was used to plot and analyze the data. For parametric data, one-way ANOVA was used to determine the variance in the group means, with Dunnett’s test used for the post hoc comparison of group means (*p* < 0.05 *, *p* < 0.01 **, *p* < 0.001 ***, and *p* < 0.0001 ****). For non-parametric analyses, a Kruskal–Wallis test was used to determine the differences in the mean rank, with a post hoc Dunn’s multiple comparisons test demonstrating significant differences between the group means (* *p* < 0.05, ** *p* < 0.01, *** *p* < 0.001, and *p* < 0.0001 ****). The cartoons in the figures were created and exported with public permission from BioRender (Toronto, ON, Canada) [54].

## 3. Results and Discussion

### 3.1. Overview of Experiments

In this study, we incorporated INI-4001, a lipidated TLR7/8 agonist, and INI-2002, a synthetic TLR4 agonist, individually or in combination into AddaS03-like emulsions (Figure 1). We qualitatively and quantitatively characterized the emulsion’s physical properties, TLR agonist recovery, and colloidal stability at refrigerated (4 °C), room (25 °C), and high (40 °C) temperatures. To determine the adjuvant activity, the emulsions were admixed with the RBD of the SARS-CoV-2 spike protein and used to immunize C57BL/6 mice. RBD-specific serum antibodies were measured to determine the humoral immunity, and an sVNT was used as a measure of the neutralizing antibody (nAB) titers. To measure the cell-mediated immune response to RBD, lymphocytes were removed from the DLNs, cultured ex vivo with antigens, and the cytokine levels in the cell supernatants were measured.

### 3.2. Qualitative Characterization of Emulsions

Since both INI-2002 and INI-4001 are amphiphilic in structure, we suspected each might have unique surfactant-like properties which could change the physical properties compared to AddaS03-like emulsions alone. Though previous TLR4 and TLR7/8 additions to O/W emulsion systems have been reported [35], systemic characterization data is often underreported, likely due to the availability of the agonists, especially TLR7/8 agonists, and the complex characterization techniques. To determine any changes in the physical properties upon the addition of INI-2002, INI-4001, or both to the AddaS03-like emulsion, cryo-EM was used to visualize any qualitative differences (Figure 2). The cryo-EM images from representative samples showed spherical particles with a high electron density that was uniform throughout the particle. The absence of any particles with distinct bilayers and the absence of amorphous aggregates suggested that O/W emulsion droplets were formed for all samples. Since the formulation of INI-4001 and INI-2002 in the DPBS buffer alone can result in large aggregates (data not shown), the lack of visible aggregates suggested that the molecules were incorporated into the emulsion.

### 3.3. Physical Characterization of Emulsions

Since quantitative differences between emulsion droplets are difficult to ascertain by cryo-EM, we measured the dose-dependent effects of TLR agonist incorporation on the hydrodynamic diameter, PDI, and zeta potential when compared to the AddaS03-like emulsion vehicle (Figure 3).

The physical characterization of AddaS03-like emulsions demonstrated a significant effect of the TLR addition on the hydrodynamic diameter (*p* = 0.04) (Figure 3A). Here, a high target concentration of INI-4001 reduced the hydrodynamic diameter of the emulsion droplets (110.6 ± 1.1) compared to that of the AddaS03-like emulsion alone (119.4 ± 2.6). Since the droplet size of squalene-based O/W emulsions has been reported to decrease with added surfactant amount [55], the INI-4001 is likely incorporated at the oil–water interface and acts as a surfactant in this system. Indeed, the hydrophilic–lipophilic balance (HLB) prediction gives a range of values (9.9–15.3) for INI-4001. While this HLB prediction corresponds to an O/W emulsifying compound on the HLB scale [48,49,50], these calculations are without respect to the compound charge and would need empirical confirmation. The addition of INI-2002 had no significant effect on the hydrodynamic diameter at these concentrations. Since software predicted an HLB value of 1.88–6.76 for INI-2002, this theory suggests that this compound had properties that were more ideal as a water-in-oil emulsion stabilizer, although again, the calculated value is limited for charged molecules.

For nanoparticle vaccines, particle size is critical to the vaccine’s efficacy [56]. Size partially controls the particle uptake by innate cells [57] and the access to resident T and B cells by trafficking to the DLN via lymphatic vessels [42]. For emulsions, MF59™ with a droplet size of 160 nm has a more potent adjuvant effect than similar emulsions with a 90 or 20 nm droplet size [58], though the reason has not been elucidated. Here, we observed that the addition of INI-4001 can reduce the emulsion droplet size by approximately 10 nm, but the small droplet size difference is unlikely to change the DLN trafficking since particles less than 200 nm in diameter access lymphatic vessels and traffic to the DLN [59]. Additionally, our unloaded AddaS03-like emulsion had a droplet size of approximately 120 nm, which is smaller than the 155 nm droplet size of AS03™ [16] and the 160 nm droplet size of the commercially available AddaS03™ [58]. Although we did not compare AddaS03-like emulsions directly to AddaS03™ or AS03™, we believe the AddaS03-like emulsion vehicle was still valid for comparing the differences due to the TLR agonist loading, and here, the physical differences compared to AS03™ and AddaS03™ may have been due to the use of reagent-grade and not clinical-grade materials, the exact processing method, or simple lab-based error.

Conversely, the TLR agonist addition did not change the PDI for the AddaS03-like emulsions (*p* = 0.23) (Figure 3B). Since the PDI ranged from 0.078–0.109 when including all emulsions tested, these emulsions were highly homogenous regardless of the addition of INI-4001 or INI-2002. As such, high-pressure homogenization should be considered sufficient for both laboratory and larger scale production. This low PDI in combination with a small droplet size also allows for filtration through a sterile membrane, which should simplify scalability and reduce the cost of production.

Lastly, the addition of TLR agonists to AddaS03-like emulsions causes significant changes in the zeta potential that are dependent on the specific agonist and its concentration (*p* < 0.0001) (Figure 3C). While the AddaS03-like emulsion gives a zeta potential of −5.2 ± 1.2 mV, a high-concentration TLR agonist addition results in −9.3 ± 0.5 mV for INI-4001, −30.5 ± 0.5 mV for INI-2002, and −28.0 ± 0.8 mV for the dual addition of INI-4001 and INI-2002. The reduction in the zeta potential can be easily explained by the formal charge of the agonists used at the formulation pH, as the formal charge of INI-4001 is −1 and the formal charge of INI-2002 is −2 (the pH range of the zeta potential measurements was found to be 6.66–6.84 here). Thus, when comparing equimolar additions of INI-2002 and INI-4001 in these emulsions, the addition of INI-2002 had a greater impact on the zeta potential. Interestingly, upon the coencapsulation of both agonists, the results demonstrated that the zeta potential is not additive, but seems to be dependent on the INI-2002 addition.

Similar to particle size, the particle surface charge has been demonstrated to affect nanoparticle vaccine efficacy [60]. Namely, the charge can dictate protein corona formulation and uptake in phagocytic cells [61], and both anionic and cationic particles have increased uptake by antigen-presenting cells compared to neutral particles [62,63]. Since a TLR agonist addition to AddaS03-like emulsions strikingly decreases the zeta potential, the addition of TLR agonists may alter the protein corona composition or the direct droplet uptake by antigen-presenting cells.

Summarily, TLR agonist addition to AddaS03-like emulsions maintains the high homogeneity of the AddaS03-like emulsion, but the TLR agonist addition can change the droplet size, and more dramatically, the zeta potential. While the emulsion droplet size, homogeneity, and zeta potential are important properties to consider for consistent formulation and manufacturing, they can also be important for biological activity.

### 3.4. TLR Agonist Recovery after Sterile Filtration

Since all tested emulsions resulted in a sterile filterable droplet size, the INI-4001 and INI-2002 compound recovery was measured by the percentage of the pre-filtration compound recovered after sterile filtration (Table 1). The INI-4001 recovery ranged from 98.8% to 101.4% when including all emulsions, excluding error, and the INI-2002 recovery ranged from 91.9% to 114% across all emulsions, excluding error. Since the recovery for each compound was greater than 90%, our processed parameters should be considered sufficient. The large variation in INI-2002 recovery was likely due to a combination of a low absorbance of the molecule at the detection wavelength and the low target amount of INI-2002, which tested the lower limits of detectability using our given protocol and detector. Additionally, clinical-grade emulsions are often made at a higher concentration and diluted down as part of the manufacturing [64] or administration, so it may be possible to improve the detection of INI-2002 by increasing the amount of all compounds used.

Regardless, the high recovery of the TLR agonists incorporated into the emulsions improves on our previously reported work with dual TLR agonist incorporation into liposomes, where we maximally recovered approximately 85% of the TLR4 agonist, CRX-601, and 78% of the lipidated TLR7/8 agonist, UM-3004, after sterile filtration [36]. In that study, the dual TLR agonist liposomes were of a similar hydrodynamic diameter (approximately 120 nm) to the emulsion droplets in this study, but the PDI was much higher for the liposomes (approximately 0.300) than that measured for the emulsions here. Consequently, we believe the increased homogeneity of the emulsion droplets resulted in a better recovery of the TLR agonists when compared to the post-filtration recovery of similar agonists from liposomes, though further particle size reduction or homogenization could improve the TLR agonist recovery in liposomes.

Crucially, the lipidation of TLR7/8 agonists has been demonstrated to allow better agonist incorporation than non-lipidated TLR7/8 agonists in lipid-based nanoformulations [65]. When incorporating the non-lipidated TLR7/8 agonist imiquimod (IMQ) and the synthetic TLR4 agonist glucopyranosyl lipid adjuvant (GLA) into neutrally charged and anionic liposomes, one group demonstrated an IMQ loading efficiency of below 10% and a further 23–45% loss of the IMQ following sterile filtration [66]. The group improved on the loading and recovery of the TLR7/8 agonist in PEGylated liposomes by replacing the IMQ with the lipidated 3M-052, which increased the total recovery of TLR7/8 to a range of approximately 70–120% [67]. Additionally, the group measured the recovery of the GLA in both studies. In neutrally charged and anionic liposomes, the group recovered the TLR4 agonist GLA at 79–81% [66], and for the PEGylated liposomes, the group recovered approximately 80–120% of the GLA after formulation [67].

While others have dually incorporated TLR4 and lipidated TLR7/8 agonists into liposomes, fewer studies have used squalene-based emulsions for dual incorporation. One group reported a dual incorporation of 3M-052 and GLA into a squalene-based emulsion [35], but the recovery of the agonists was not measured. In a similar emulsion, the same group singly incorporated 3M-052 and reported 100% maximal recovery, though variations in the manufacturing process and agonist concentration could reduce the recovery to below 40% [68] (p. 7).

Summarily, the high recovery of INI-4001 and INI-2002, singly and in combination, compares favorably to the recovery of TLR4 and TLR7/8 agonists from other lipid-based nanoformulations, including liposomes and squalene-based emulsions. Furthermore, these AddaS03-like emulsions with the addition of TLR4 and TLR7/8 alone or in combination are viable formulations for larger scale production with minor modifications to the detection procedures.

### 3.5. Colloidal Stability of Emulsions in Storage

In order to measure the colloidal stability of AddaS03-like emulsions containing low and high concentrations of INI-2002, INI-4001, or combinations of the two, the emulsions were stored at 4 °C, 25 °C, and 40 °C to resemble refrigerated, room temperature, or high temperature storage, respectively. The results demonstrated no changes to the droplet size, polydispersity, or zeta potential over the course of the experiment at 4 °C or 25 °C (Figure 4). Thus, the addition of INI-2002 and INI-4001, alone or in combination, had no observable effect on the colloidal stability when compared over time. While the duration of this study was 3 months, AS03™ is highly stable for up to 10 months at 4 °C [53] and resistant to temperature fluctuations [16]. Since the addition of the TLR agonists resulted in minimal changes to the hydrodynamic diameter and PDI, we expect the colloidal stability to be similar. Additionally, previous results indicate that emulsions with a higher magnitude of zeta potential are more colloidally stable due to the charge repulsion of droplets [69]. Informally, we observed no change in the droplet size, polydispersity, or zeta potential for up to 6 months at room temperature and 9 months for the refrigerated emulsions (results not shown).

In the context of SARS-CoV-2 vaccines, we believe the colloidal stability of squalene-based emulsions, with or without the addition of TLR agonists, offers a stark advantage over the current mRNA-based vaccines. While Moderna’s mRNA-1273 claims up to 30-day stability at 2–8 °C, its stability drops to 12 h at room temperature [70]. Worse, BioNTech-Pfizer’s BNT162b2 claims 30-day stability at 2–8 °C, 2- to 6-h stability at room temperature, and requires freezing at −80 to −60 °C for longer-term storage [71]. Conversely, AS03™ has been demonstrated to effectively seroconvert greater than 95% of patients after the adjuvant was stored for 2.5 years and the antigen for 4 years [72]. In this case, the robust colloidal stability of AS03™ simplifies shipping and storage, and lowers the cost by reducing the reliance on complex cold chain logistics when compared to the current mRNA vaccines. Additionally, the production of the adjuvant and the recombinant protein antigen can be decoupled, so a surfeit of adjuvant could be stockpiled [72]. Since squalene-based vaccines have demonstrated cross-clade protection for influenza in animal studies [73], stockpiling vaccine adjuvants is an enticing strategy that could be applied to the emerging SARS-CoV-2 variants.

### 3.6. Humoral Response of Mice Immunized with AddaS03-Like Emulsions Containing TLR Agonists

To determine the effect of the addition of TLR4 or TLR7/8 agonists in AddaS03-like emulsions, INI-2002 or INI-4001 were incorporated singly or together, and the emulsions were tested for their ability to adjuvant the immune response to recombinant RBD in vivo in C57BL/6 mice (Figure 5A). To determine the TLR-mediated differences in the resulting humoral immune response, the total RBD-specific IgG titers and the titers of RBD-specific IgG1 and IgG2c were measured in the serum after the booster immunization (Figure 5B–D). AddaS03-like and MF59-like emulsions were also included to approximate the FDA-approved emulsions, AS03™ and MF59™.

We measured a significant increase in the RBD-specific total IgG titers when the RBD antigen was adjuvanted with squalene-based emulsions compared to vaccinating with the RBD antigen alone (*p* < 0.0001). Though increases in the total IgG were seen with all emulsions, only the higher dose of INI-2002 significantly increased the total IgG production when compared to AddaS03-like and MF59-like emulsions (Figure 5A). When compared directly, this equated to a 22.9-fold increase when compared to the AddaS03-like formulation, and 16-fold increase when compared to MF59-like emulsion, while controlling for the squalene dose (Appendix A). All other TLR additions increased the total IgG by 1.3–2.7-fold when compared to the AddaS03-like control and 0.9–1.7-fold when compared to the MF59-like control. Here, our results confirmed that the AddaS03-like and MF59-like emulsions were able to robustly increase the total IgG titers, and a higher dose of the INI-2002 addition can further increase that response. Therefore, the addition of INI-4001, INI-2002, or combinations thereof to AddaS03-like emulsions was at least equivalent to the AddaS03-like emulsion vehicle alone for eliciting the total RBD-specific IgG. Since AS03™ robustly increases an effective humoral response in the current SARS-CoV-2 vaccine candidates [17], maintaining a similar or improved total IgG response upon TLR addition is a worthy benchmark.

The IgG isotypes were also measured as an indication of the type of immune response being generated. We observed a significant increase in the RBD-specific IgG1 titers when RBD was adjuvanted with squalene-based emulsions *(p* = 0.023), but we did not observe significant differences when the emulsions containing TLR agonists were compared to the AddaS03-like or MF59-like controls (Figure 5C). Adding TLRs to AddaS03-like formulations resulted in a range of differences, from 0.5- to 4.9-fold compared to the AddaS03-like formulations and from 0.4- to 4.3-fold compared to the MF59-like formulations (Appendix A). In contrast to IgG1, we observed significant increases in the IgG2c titers when adding a higher dose or combination of TLR agonists to AddaS03-like emulsions (*p* < 0.0001, Figure 5D). For the high dose of INI-4001, we observed a near 110-fold increase in the IgG2c titers when compared to the AddaS03-like emulsion and a 62-fold increase when compared to the MF59-like emulsion (Appendix A). For the high-dose INI-2002 emulsion, we observed approximately 962-fold and 542-fold increases in the IgG2c titers compared to the AddaS03-like and MF59-like emulsions, respectively.

The full effect of these adjuvants on the balance between IgG1 and IgG2c isotype production becomes most readily apparent when comparing the ratios of IgG2c/IgG1 mean antibody titers for each group (Figure 5E). This ratio shows that the higher doses of INI-2002 and INI-4001, singly and in combination, effectively skewed the isotype balance towards IgG2c, which is suggestive of a Th1-type biasing of the immune response [73]. This is in stark contrast to the AddaS03-like and MF59-like emulsions alone.

### 3.7. AddaS03-Like Emulsions Effectively Neutralize ACE2 Binding to the Spike Protein in SARS-CoV-2 Variants

While a definitive correlate of protection against SARS-CoV-2 has yet to be codified [74], both spike-specific and RBD-specific IgG have been reported to correlate with virus neutralization in convalescent plasma from recovered patients [75], and virus neutralizing antibodies (nAbs) have been reported to be highly predictive of immune protection in individuals with symptomatic SARS-CoV-2 [76]. Since we observed increases in the RBD-specific antibody titers when adjuvating RBD with squalene-based emulsions, we sought to determine if the increase in titers also leads to an increase in nAbs. Here, we used a sVNT that measures the ability of serum antibodies to block the binding of ACE2 to RBD in order to determine the neutralization potential of the antibodies generated by immunization (Figure 6).

The sera from mice adjuvanted with squalene-based emulsions more effectively inhibited the binding of ACE2 to SARS-CoV-2 RBD (A/Wuhan), the spike protein (A/Wuhan), the spike protein containing the common D614G mutation, and the spike protein from the Alpha variant (B.1.1.7) compared to the sera from mice immunized with RBD alone (*p* < 0.0001 for each). Specifically, AddaS03-like emulsions resulted in percentages of binding inhibition of 97.5% for the A/Wuhan RBD, 95.3% for the A/Wuhan spike protein, 97.5% for the D614G spike protein, and 97.5% for the Alpha spike protein. The TLR agonist addition to AddaS03-like emulsions did not significantly increase these figures. For the Beta (B.1.351) and Gamma (P.1) variants of concern, the AddaS03-like emulsion provided 89.0% and 86.3% inhibition of binding to the spike protein, though neither was significantly higher when compared to the mice immunized with RBD alone. Similarly, the TLR addition did not increase these figures.

Taken together, the results from the sVNT reflect previously published data, where SARS-CoV-2 vaccines adjuvanted with AS03™ increased the production of nAbs. Specifically, vaccination with AddaS03™, the commercially available emulsion based on AS03™, and RBD increased the nAb titers in the sera of mice, which then effectively reduced the infection by a SARS-CoV-2 reference strain, the Alpha variant, and, to a lesser extent, the Beta variant of SARS-CoV-2 in vitro [77]. In non-human primates, vaccination with recombinant spike proteins adjuvanted with AS03™ effectively controlled SARS-CoV-2 replication in the airways, and the resulting serum IgG protected against SARS-CoV-2 challenge when adoptively transferred to hamsters [78]. Most importantly, the results from phase I/II in humans demonstrates that AS03™ mediated a 4-fold increase in the nAb titers in approximately 75% of patients after two vaccinations with AS03™ and the spike antigen when compared to the baseline titers before vaccination [17]. While the duration of protection for AS03™-induced SARS-CoV-2 nAbs remains to be studied, the drop in neutralization that we measured against the Beta and Gamma variants compared to the A/Wuhan strain echoes a reduction in the neutralization to the same variants of concern by the human convalescent serum [11].

While it is clear that the AddaS03-like emulsion increased the inhibition of ACE2 binding to RBD, the role of adding TLR agonists, singly or in combination, to squalene-based emulsions remain unclear. Our TLR agonist additions to AddaS03-like emulsions resulted in high ACE2 binding inhibition, but their equivalence to the AddaS03-like emulsion suggests that TLR agonist addition is not necessary for nAb production here. Previous results using delivery vehicles other than emulsions suggested that a TLR4 or TLR7/8 agonist, alone or in combination, produced robust nAb titers [79,80,81,82,83]. The TLR4 agonist Monophosphoryl Lipid A demonstrated a production of effective nAb titers comparable to that of AddaVax when used to adjuvant recombinant RBD [79], though combinations of the two were not tested. In another study, an imidazoquinoline TLR7/8 agonist linked to an amphiphile, induced nAbs that more effectively neutralized SARS-CoV-2 than AddaVax, each after one dose [80]. The vaccine formulation Covaxin, which chemisorbs an imidazoquinoline TLR7/8 agonist to particulate aluminum hydroxide, demonstrated efficacious neutralizing activity against SARS-CoV-2 variants in humans [81,82] and as a result, the World Health Organization issued the vaccine for emergency use [84]. Lastly, a liposome formulation with a dual TLR4 and TLR7/8 agonist combination protected humanized ACE2 mice from lethal infection by SARS-CoV-2 while also inducing nAb production [83].

### 3.8. Cell-Mediated Response of Mice Immunized with AddaS03-Like Emulsions Containing TLR Agonists

To more formally characterize the adjuvant effect of TLR agonists incorporated in squalene-based emulsions on cell-mediated immunity, DLNs were collected after booster immunization, the cell suspension was cultured with RBD to restimulate antigen-specific T cells, and the T cell cytokines IFNγ, IL-17A, and IL-5 were measured as an indication of the type of effector T cells that were present (Figure 7).

The production of IFNγ is considered the hallmark of Th1 CD4^+^ T cells, and cell supernatants from mice that received the higher dose of INI-4001 contained a significantly increased concentration of IFNγ compared to those adjuvanted with AddaS03-like or MF59-like emulsions alone (Figure 7A). When comparing means, this was a 9.8-fold and 8.9-fold increase compared to AddaS03-like and MF59-like controls, respectively (Appendix A). We also observed increases in the mean IFNγ production from the INI-4001 and INI-2002 coencapsulated groups that equated to a range of a 3.3–4.7-fold increase compared to the AddaS03-like or MF59-like emulsions. A Th1-biased response is considered the ideal anti-viral immune response, with IFNγ playing an essential role in the development of CD8^+^ T cells [85]. Here, we observed that the addition of INI-4001 to AddaS03-like emulsions was an effective way to increase the IFNγ production, which provided evidence of a Th1-type cell-mediated response. These results also corroborated the ability of INI-4001 to selectively increase the levels of IgG2c antibodies, which are associated with Th1 responses. While IFNγ has not been formally defined as a correlate of protection for SARS-CoV-2 in vaccine studies, it is likely to play an important role due to its essential function in immunity to other viral diseases, including influenza [86]. Interestingly, researchers reported SARS-CoV-2’s ability to dampen the host’s IFNγ response in early infection [87], and additionally, patients with loss-of-function variants of the TLR7 receptor showed a blunted IFNγ response that correlated with enhanced COVID-19 [88].

While the increased production of IgG2c correlated with a Th1 response for the groups receiving INI-4001, the production of IFNγ was not seen in the groups that received the TLR4 agonist INI-2002, despite a similar increase in IgG2c levels. Th17 cells compose another subset of effector CD4^+^ T cells, which secrete IL-17 and are largely known for their role in the fight against mucosal bacterial and fungal infections [89,90]. The emulsions containing INI-2002 resulted in increased levels of IL-17A (Figure 7B), with the lower dose showing a 5.2-fold increase when compared to the AddaS03-like emulsion and a 14.1-fold increase when compared to the MF59-like emulsion (Appendix A). Although the impact of eliciting Th17 cells in vaccination for COVID-19 is unknown, IL-17A is implicated in the development of severe acute respiratory distress syndrome (ARDS) [91]. Additionally, an IL-17A-blocking monoclonal antibody was shown to protect mice from a COVID-19-induced cytokine storm and the resulting lung inflammation [92]. As such, the result of an increase in IL-17A for INI-2002 when added to AddaS03-like emulsions may warrant further study, although these results suggested an inverse effect of dose, and the addition of the TLR7/8 agonist resulted in decreased IL-17 production.

Th2 cells provide immunity to extracellular organisms by promoting antibody production over the development of cell-mediated responses through cytokine secretion. We used levels of IL-5 in the cell supernatants as a signature of Th2 development. As expected, AddaS03-like and MF59-like emulsions drove high levels of RBD-specific IL-5 release, consistent with the Th2-dominated response seen after an AS03-adjuvanted vaccination against the spike protein in the rhesus macaque model [93]. Strikingly, we observed dose-dependent drops in the IL-5 production when INI-4001 and/or INI-2002 were incorporated into the AddaS03-like emulsions (Figure 7c and Appendix A). This drop in IL-5 production correlated with an increased IgG2c/IgG1 ratio (Figure 5E), and supported the hypothesis that the addition of TLR4 or TLR7/8 agonists drives the immune response to AddaS03-like adjuvanted vaccines away from a Th2 response. Since research with SARS-CoV-1 and MERS-CoV-1 has suggested that Th2-type responses may exacerbate lung inflammation [94], we see this reduction in IL-5 production as an advantage for vaccination against SARS-CoV-2.

The use of dual TLR4 and TLR7/8 agonism to develop a Th1-biased response has been described previously in liposomes [35,36,83], but our result is the first we know of that demonstrates the effect in squalene-based emulsions. GLA, a TLR4 agonist, and 3M-052, a lipidated TLR7/8 agonist, were successfully coencapsulated in a squalene-based emulsion, but the result was a more balanced Th1/Th2 response [35]. While the dual INI-4001 and INI-2002 addition skewed the response most effectively, INI-4001 or INI-2002 alone was sufficient to substantially reduce the Th2 development. Th1 skewing by a single TLR4 agonist in a squalene-based emulsion, GLA-SE, has also been previously described [95].

Overall, Th1-type responses have been correlated with improved outcomes for patients infected with SARS-CoV-2 [96]. While both AS03™ and MF59™ increase spike-specific IgG titers in humans [17,97], neither has been demonstrated to produce durable Th1-skewed responses. Importantly, AS03™ produced a Th0/Th2-dominated response to the spike antigen in non-human primates [93] and a largely Th2-dominated profile in influenza vaccination [98]. Here, we demonstrate that the addition of TLR agonists to AddaS03-like emulsions is an effective strategy to bias toward a Th1-type humoral response. While this strategy may be successful for other squalene-based emulsions and other TLR agonists, it will likely depend on specific factors such as the TLR4 and/or TLR7/8 agonist activity, relative dosages, antigens used, and model systems.

## 4. Conclusions

While currently approved mRNA vaccines have been crucial to reducing the mortality and morbidity of SARS-CoV-2, newer vaccine approaches, including adjuvanted recombinant protein subunit vaccines, offer promise in the continued fight against SARS-CoV-2. Squalene-based O/W emulsions, including those further adjuvanted with TLR4 and/or lipidated TLR7/8 agonists, alone or in combination, can effectively combine the desirable stability characteristics of emulsions with Th1-biasing immune responses. Here, our results suggest that the incorporation of INI-4001 and INI-2002, alone or in combination, into AddaS03-like emulsions results in minimal change in the physical properties and retains the stability of AddaS03-like emulsions upon storage at refrigeration or room temperatures. Additionally, the addition of TLR4 or TLR7/8 agonists can dramatically change the adaptive immune response induced by AddaS03-like emulsions by skewing the emulsions away from a Th2-dominated response. Lastly, the addition of these TLR agonists provided serum neutralization to emerging SARS-CoV-2 variants that was equivalent to AddaS03-like emulsions alone. Thus, using TLR4 or lipidated TLR7/8 agonists to adjuvant squalene-based O/W subunit vaccines provides another strategy to fight SARS-CoV-2 variant strains, especially in developing countries where the cost and storage are crucial factors in the fight.

## Figures and Tables

**Figure 1 pharmaceutics-14-01455-f001:**
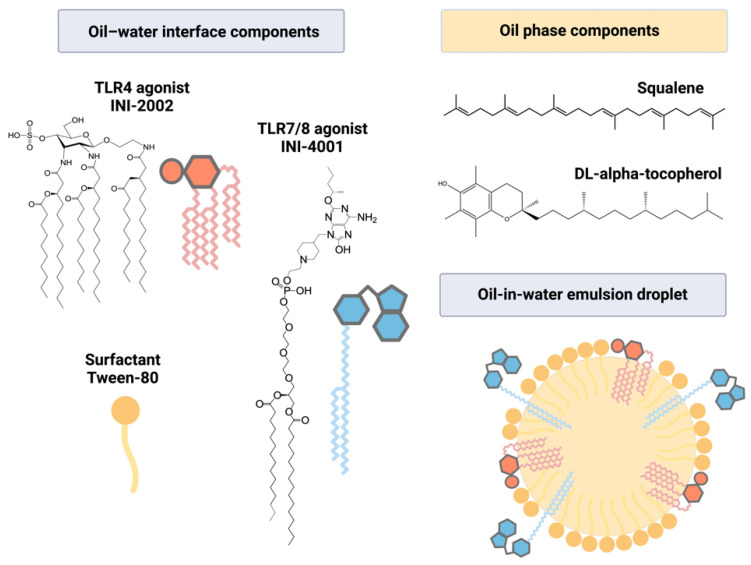
Structures of compounds incorporated in a squalene-based emulsion. INI-4001, a lipidated oxoadenine and TLR7/8 agonist, and/or INI-2002, a synthetic TLR4 agonist, were incorporated into an AddaS03-like emulsion. Lipid moieties present in the compounds allowed incorporation into the oil–water interface, along with the surfactant Tween-80. The oil phase was comprised of squalene and DL-α-tocopherol.

**Figure 2 pharmaceutics-14-01455-f002:**
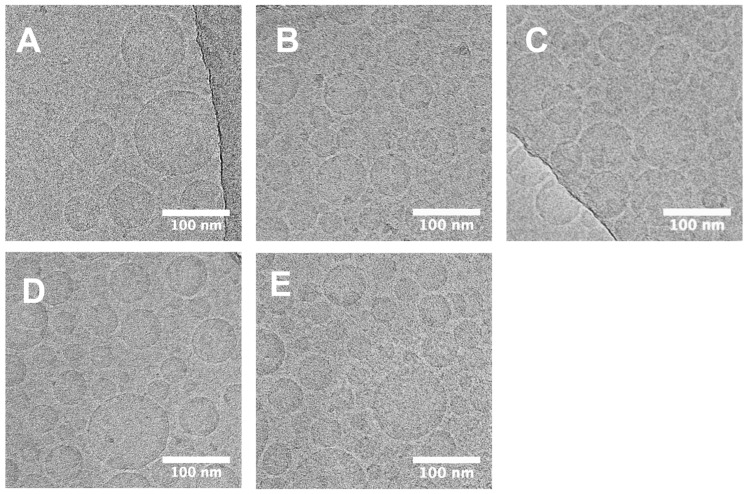
Cryo-EM images show no qualitative difference between AddaS03-like emulsions and those containing INI-4001 or INI-2002 alone or in combination. Cryo-EM was used to determine qualitative differences between (**A**) AddaS03-like emulsions and AddaS03-like emulsions containing (**B**) INI-4001 at 1.75 mM, (**C**) INI-2002 at 1.75 mM, and combinations of INI-4001 and INI-2002 at (**D**) 1.75 mM and 0.175 mM, respectively, and (**E**) 1.75 mM and 1.75 mM, respectively. Images were taken at 22,000× magnification with scale bars indicating 100 nm. Images are representative of each emulsion.

**Figure 3 pharmaceutics-14-01455-f003:**
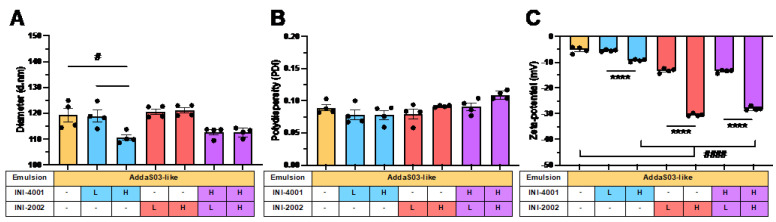
Effect of incorporation of INI-4001 and INI-2002 on hydrodynamic diameter, polydispersity (PDI), and zeta potential in AddaS03-like emulsions. INI-4001 or/and INI-2002 were incorporated alone or in combination into AddaS03-like emulsions at “low” (L) 0.175 mM or “high” (H) 1.75 mM target concentrations. Dynamic light scattering was used to measure (**A**) hydrodynamic diameter (Z.Avg, nm), (**B**) PDI, and (**C**) zeta potential (mV) at a pH of 6.7–6.8. Physical properties are given as mean ± SEM with black dots representing each replicate emulsion (*n* = 4). A one-way ANOVA with a post hoc Tukey’s test for multiple comparisons demonstrated significant differences between group means within compound concentrations (**** *p* < 0.0001) or when compared or to the AddaS03-like vehicle (# *p* < 0.05, #### *p* < 0.0001).

**Figure 4 pharmaceutics-14-01455-f004:**
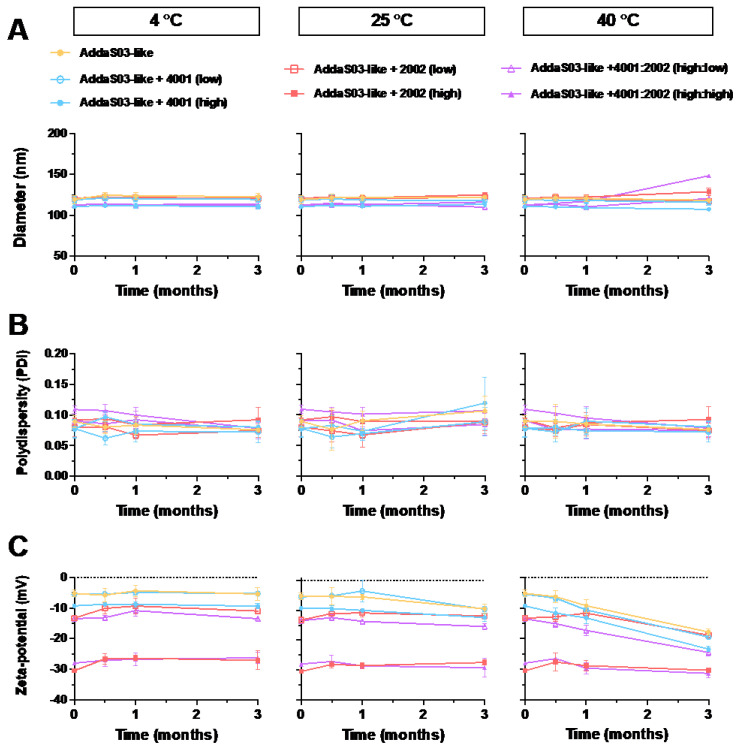
AddaS03-like emulsions maintained high colloidal stability at various temperatures with the addition of TLR4 and/or lipidated TLR7/8 agonists. An AddaS03-like emulsion was used to incorporate INI-4001 (TLR7/8 agonist) and/or INI-2002 (TLR4 agonist). Agonists were incorporated at low (0.175 mM) or high (1.75 mM) target concentrations, singly or coencapsulated. Dynamic light scattering was used to measure (**A**) hydrodynamic diameter (Z.Avg, nm), (**B**) polydispersity (PDI), and (**C**) zeta potential (mV) after storage at 4, 25, or 40 °C. Physical properties are given as mean ± SEM for replicate emulsions (*n* = 4).

**Figure 5 pharmaceutics-14-01455-f005:**
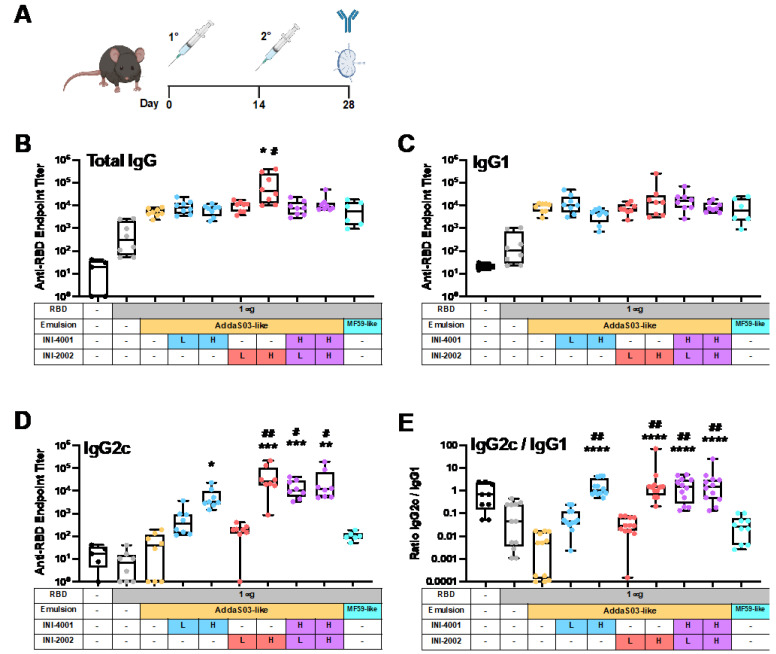
Addition of TLR4 and/or TLR7/8 agonists to AddaS03-like emulsions increases RBD-specific IgG2c antibody titers without reducing total IgG or IgG1 titers. (**A**) C57BL/6 mice were immunized twice with recombinant SARS-CoV-2 RBD admixed with emulsions containing INI-4001 (TLR7/8 agonist) or INI-2002 (TLR4 agonist), singly or dually encapsulated. Agonists were administered at a low dose (L, 0.876 nmol) or high dose (H, 8.76 nmol). Fourteen days after the booster immunization, anti-RBD serum antibody titers were measured for (**B**) total IgG, (**C**) IgG1, (**D**) IgG2c, and (**E**) the ratio of IgG2c/IgG1. Results are presented as box-and whisker plots representing the group median with quartiles (box) and range (whiskers) overlaid with data from individual animals. A Kruskal–Wallis test with a post hoc Dunn’s multiple comparisons test was used to determine significant differences in group mean rank when compared to AddaS03-like (* *p* < 0.05, ** *p* < 0.01, *** *p* < 0.001, and **** *p* < 0.0001) or MF59-like (# *p* < 0.05, ## *p* < 0.01) emulsions alone.

**Figure 6 pharmaceutics-14-01455-f006:**
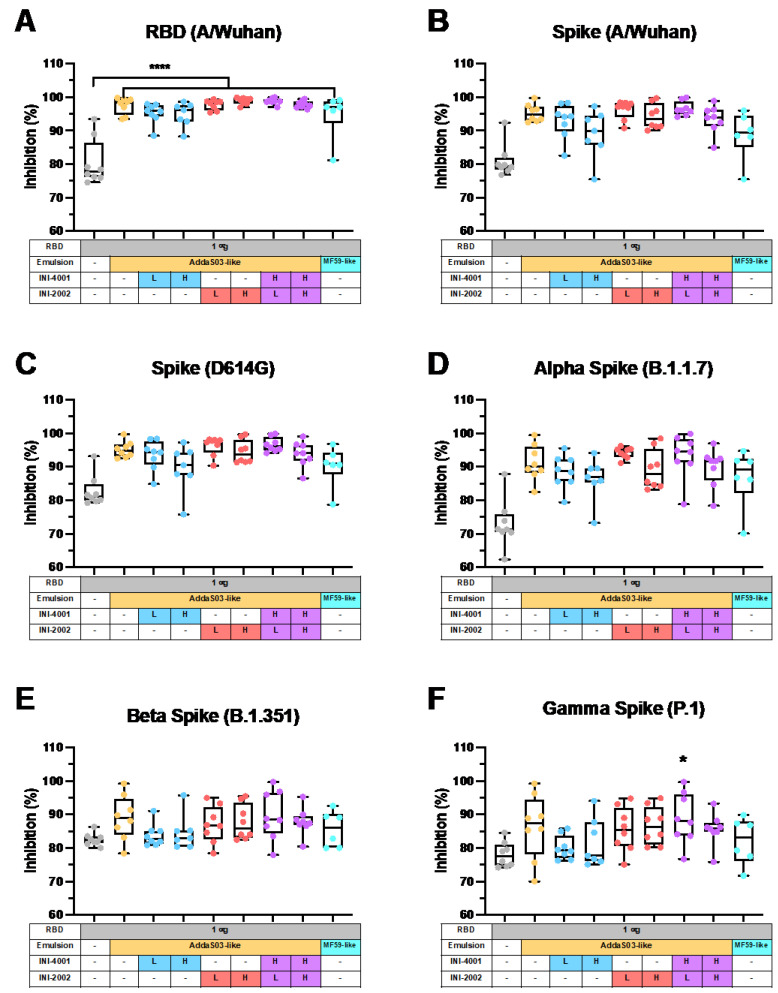
AddaS03-like emulsions, with or without toll-like receptor (TLR) agonists, result in neutralization of SARS-CoV-2 variants. A surrogate virus neutralization test (sVNT) was used to test the neutralization potential of serum taken fourteen days after the booster immunization. The percent reduction in the binding of recombinant ACE2 to the indicated SARS-CoV-2 protein is given for (**A**) RBD (A/Wuhan), (**B**) spike (A/Wuhan), (**C**) spike (D614G mutation), and spike from the variants of concern: (**D**) Alpha (B.1.1.7), (**E**) Beta (B.1.351), and (**F**) Gamma (P.1). Results are presented as box-and whisker plots representing the median and quartiles (box) and range (whiskers) overlaid with data from individual animals. A one-way ANOVA with a post hoc Tukey’s test for multiple comparisons demonstrated significant differences between group means (* *p* < 0.05 and **** *p* < 0.0001) when compared to unadjuvanted RBD (*n* = 6–8).

**Figure 7 pharmaceutics-14-01455-f007:**
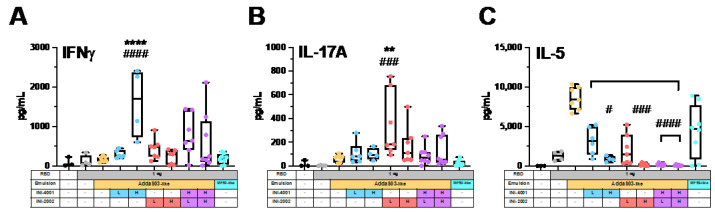
Addition of TLR4 or TLR7/8 agonists to AddaS03-like emulsions significantly skewed away from Th2-type cytokine profiles in the draining lymph node (DLN). Fourteen days post-secondary immunization, lymphocytes were isolated from the DLN, grown ex vivo, and stimulated with SARS-CoV-2 RBD for 72 h. RBD-specific (**A**) IFNγ, (**B**) IL-17A, and (**C**) IL-5 were measured in the cell supernatant for each group. A one-way ANOVA with a post hoc Tukey’s test for multiple comparisons demonstrated significant differences in group means when compared to the AddaS03-like (** *p* < 0.01, **** *p* < 0.0001) or MF59-like (# *p* < 0.05, ### *p* < 0.001, and #### *p* < 0.0001) emulsions alone.

**Table 1 pharmaceutics-14-01455-t001:** TLR agonist recovery after sterile filtration of emulsions. INI-4001 and INI-2002 were incorporated into AddaS03-like emulsions singly or in combination at given target concentrations. Percentage recovery was calculated by comparing the agonist concentration pre- and post-filtration with a 0.22 μm filter. Results are presented as mean ± SEM for replicate emulsions (*n* = 4).

Emulsion	Target Concentration INI-4001 (mM)	Target Concentration INI-2002 (mM)	INI-4001 Recovery (%)	INI-2002 Recover (%)
AddaS03-like + 4001	0.175	-	100.2 ± 2.2	
	1.75	-	101.4 ± 1.2	
AddaS03-like + 2002		0.175		114.4 ± 17.9
		1.75		108.2 ± 5.6
AddaS03-like + 4001:2002	1.75	0.175	99.8 ± 1.3	92.9 ± 4.1
	1.75	1.75	98.8 ± 1.3	91.9 ± 2.8

## Data Availability

The data presented in this study are available from the corresponding author upon request.

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
