# Peer review of "Using Dual Toll-like Receptor Agonism to Drive Th1-Biased Response in a Squalene- and α-Tocopherol-Containing Emulsion for a More Effective SARS-CoV-2 Vaccine"

_pharmaceutics, 2022, doi:10.3390/pharmaceutics14071455_

Round 1

Reviewer 1 Report

1. Draw the structures of squalene and alpha-tocopherol in Fig.1.

2. Improve the contrast of Cryo-TEm images in Fig.2.

3. Check the typo in the temperature in Fig.4.

4. Label  X axis of all graphs. Because there are too many ticks,it is difficult to recognize only by a distant labels.

Author Response

Reviewer 1

  1. Draw the structures of squalene and alpha-tocopherol in Fig.1.

We have modified Fig. 1 to include the structures of squalene and alpha-tocopherol.

  1. Improve the contrast of Cryo-TEM images in Fig.2.

We have modified the Fig. 2 to improve the contrast of emulsion droplets. We believe this slightly improved the figure but were limited by the resolution of the images when using linear image modifications (brightness and contrast).

  1. Check the typo in the temperature in Fig.4.

We fixed the “°C” within the Figure 4 (Line 484). We also checked the error on a different computer. If the error persists due to file corruption, please notfiy our group.

  1. Label  X axis of all graphs. Because there are too many ticks, it is difficult to recognize only by a distant labels.

To be clearer to the reader, x-axis labels were added to Figures 4, 5, and 6.

Submission Date

04 June 2022

Date of this review

10 Jun 2022 02:45:16

Reviewer 2 Report

The manuscript by Short et al. describes two vaccine prototypes against SARS-Cov-2 with improved Th1-type response, designed using two TLRs agonist and a AS03-like nanoemulsion as adjuvant and antigen carriers.

The work described in this manuscript is of good technical quality and is thoroughly executed and well-presented. The experiments performed are appropriated to the aim of the work and the discussion and conclusions are properly argued.

There are minor changes that could improve the quality of the work.

1.      The relevance of Th1-biased immune response for a Covid-19 vaccine is not clearly stated in the manuscript. 

2.      Lines 327-332 “We suspect this difference in our preparations is due to the use of reagent grade and not clinical grade materials, the former of which may contain less defined impurities and oxidation productions. Though the difference in hydrodynamic diameter is approximately 40 nm, emulsion droplet sizes of 90 nm and 20 nm have been reported to reduce activity in MF59-like emulsions compared to the 160 nm sized MF59™ [41], [42], so we do not suspect this difference to alter pharmacokinetics.”

The authors argue that the difference in size found between their AS03-like emulsion and the commercial one rely on the purity of the reagents. However, there are other relevant parameters, such as the intensity or the time of the ultra-acoustic focusing, or the conditions used for the high-pressure homogenization. Where these conditions equal or similar to the ones used in the commercial formulation? It would be also helpful to add the purity of the reagents, if known.

On the other hand, the comparison with the MF59TM emulsion in this paragraph is confusing. Please rewrite that sentence in a more comprehensive manner.

3.      In lines 608-610 the authors discussed the IL-5 levels as: “Since research with SARS-608 CoV-1 and MERS-CoV-1 has suggested that Th2-type responses may exacerbate lung inflammation [68], we see this reduction of IL-5 production as an advantage for vaccination against SARS-CoV-2.”

The Th2 immune response is needed to obtain the neutralizing antibodies, hence, the production of IL-5 in a vaccine against SARS-Cov2 is a positive outcome. However, very high levels could contribute to the cytokine storm in the lung, which is a different process.

The reduction of IL-5 levels induced by the emulsions in comparison to the adjuvant alone is very likely due to the INFg production, which inhibits the Th2-type cytokines and induces the more balanced Th1/Th2 immune response observed with the adjuvanted emulsions.

Minor errors:

Table 1. Table title is too long and descriptive. Consider using table footnotes for descriptive data and definitions, and concise titles.

Figure 4. The symbol of degrees Celsius is wrong. The legend fragmented into three graphs is confusing at first glance.

Figure 5. The symbol of micrograms is wrong. Which does p<0.0001 in the graph refer to? RBD alone?

Ref. 36 is incomplete.

Author Response

Reviewer 2

  1. The relevance of Th1-biased immune response for a Covid-19 vaccine is not clearly stated in the manuscript. 

We thank the reviewer for pointing out that we’ve failed to sufficiently address this point, as it is central to our vaccination strategy. We have added information to the introduction that gives evidence for the importance of T cell-mediated immunity, and particularly Th1 and CD8 T cell responses against SARS-CoV-2 and other respiratory viruses (Lines 53-92).

  1. Lines 327-332 “We suspect this difference in our preparations is due to the use of reagent grade and not clinical grade materials, the former of which may contain less defined impurities and oxidation productions. Though the difference in hydrodynamic diameter is approximately 40 nm, emulsion droplet sizes of 90 nm and 20 nm have been reported to reduce activity in MF59-like emulsions compared to the 160 nm sized MF59™ [41], [42], so we do not suspect this difference to alter pharmacokinetics.”

The authors argue that the difference in size found between their AS03-like emulsion and the commercial one rely on the purity of the reagents. However, there are other relevant parameters, such as the intensity or the time of the ultra-acoustic focusing, or the conditions used for the high-pressure homogenization. Where these conditions equal or similar to the ones used in the commercial formulation?

In our hands, high-pressure homogenization (HPH) reduces particle size and polydispersity much more than ultra-acoustic focusing (UAF), so we use UAF to make a crude emulsion and HPH to process the crude emulsion. For HPH, we use conditions similar to other squalene-based emulsions using the technique. Here, we use 25,000 PSI for 6 passes on a Microfluidics LV1. For AddaS03™ and AS03™, the exact processing conditions are difficult to determine due to trademark/trade secrets, but for the FDA approved emulsion MF59™, 5 passes at 12,000 PSI are sufficient for size reduction on a Microfluidics M110Y (Ott, Barchfeld, and Nest 1995; Fang and Hora 2000). Additionally, for the TLR4 agonist containing emulsion GLA-SE, 12 passes at 30,000 PSI on a Microfluidics M110P is sufficient processing (Fox et al. 2013). Note that the Microfluidics M110Y and M110P are larger machines used for HPH with similar processing geometry to the LV1 but larger volume. To be clear to the reader, in the section “2.2 Emulsion selection and preparation” we added that our process conditions are similar to other clinically relevant emulsions and cited these three examples (Lines 140-141) .

Fang, Jia-Hwa, and Maninder Hora. 2000. “The Adjuvant MF59: A 10-Year Perspective Gary Ott, Ramachandran Radhakrishnan,.” In Vaccine Adjuvants: Preparation Methods and Research Protocols, edited by Derek T. O’Hagan, 211–28. Methods in Molecular MedicineTM. Totowa, NJ: Springer New York. https://doi.org/10.1385/1-59259-083-7:211.

Fox, Christopher B., Magdalini Moutaftsi, Julie Vergara, Anthony L. Desbien, Ghislain I. Nana, Thomas S. Vedvick, Rhea N. Coler, and Steven G. Reed. 2013. “TLR4 Ligand Formulation Causes Distinct Effects on Antigen-Specific Cell-Mediated and Humoral Immune Responses.” Vaccine 31 (49): 5848–55. https://doi.org/10.1016/j.vaccine.2013.09.069.

Ott, Gary, Gail L. Barchfeld, and Gary Van Nest. 1995. “Enhancement of Humoral Response against Human Influenza Vaccine with the Simple Submicron Oil/Water Emulsion Adjuvant MF59.” Vaccine 13 (16): 1557–62. https://doi.org/10.1016/0264-410X(95)00089-J.

It would be also helpful to add the purity of the reagents, if known.

To address this, we added a section “2.1 Materials” that includes reagent grades and vendor information for ingredients used to make the emulsions (Lines 107-113). Overall, synthesized reagents were of 99% plus purity, vendor materials for emulsions were research grade, and HPLC reagents were analytical grade. Additionally, we added vendor information to reagents and equipment throughout the “Materials and Methods” section.

On the other hand, the comparison with the MF59TM emulsion in this paragraph is confusing. Please rewrite that sentence in a more comprehensive manner.

We have reviewed the paragraph and agree that it is confusing. Our overall point is TLR agonist addition to AddaS03-like emulsions changes emulsion droplet size (minimally) and zeta-potential (strikingly), and size and zeta-potential are also linked to adjuvant activity. To highlight this, we added more comprehensive discussion of the biological effects of droplet size (Lines 357-371) and zeta-potential (Lines 391-402).

  1. In lines 608-610 the authors discussed the IL-5 levels as: “Since research with SARS-608 CoV-1 and MERS-CoV-1 has suggested that Th2-type responses may exacerbate lung inflammation [68], we see this reduction of IL-5 production as an advantage for vaccination against SARS-CoV-2.”

The Th2 immune response is needed to obtain the neutralizing antibodies, hence, the production of IL-5 in a vaccine against SARS-Cov2 is a positive outcome. However, very high levels could contribute to the cytokine storm in the lung, which is a different process.

The reduction of IL-5 levels induced by the emulsions in comparison to the adjuvant alone is very likely due to the INFg production, which inhibits the Th2-type cytokines and induces the more balanced Th1/Th2 immune response observed with the adjuvanted emulsions.

We appreciate the reviewer’s argument that a Th2-mediated immune response tends to correlate with higher neutralizing antibody titers. However, a Th2-skewed response is not required for the development of RBD-specific antibodies, and a Th1 response does not preclude their development. Development of T follicular helper cells is independent of either Th1 or Th2 cell development (Crotty 2014). It is true that the development of Th1 cells in place of Th2 can result in a lower level of T cell help to B cells, as the Th1 cells will be unable to provide the additional help that Th2 cells would, and therefore leave the Tfh cells as the only source. However, a good antibody response still may develop, and indeed we see that the combination of INI-4001 and lower dose of INI-2002 produces the best antibody-mediated pseudoneutralization result shown in Figure 7, while the T cell assay results in no detectable IL-5 as a result of this formulation. We envision the ideal vaccine as one that develops both good neutralizing antibodies and at the same time a Th1-biased cellular immune response to provide CD8+ T cell help while potentially guarding against the development of Th2-driven immunopathology.

Crotty, Shane. 2014. “T Follicular Helper Cell Differentiation, Function, and Roles in Disease.” Immunity 41 (4): 529–42. https://doi.org/10.1016/j.immuni.2014.10.004.

Minor errors:

Table 1. Table title is too long and descriptive. Consider using table footnotes for descriptive data and definitions, and concise titles.

We have shortened the title of Table 1 and edited the legend to be more concise (Lines 418-421). Additionally, we decided to add two columns to Table 1 for target concentrations of the TLR agonists. This allowed us to reduce the total words in the table legend even further. We believe the changes give the title clarity without being verbose.

Figure 4. The symbol of degrees Celsius is wrong. The legend fragmented into three graphs is confusing at first glance.

We fixed the incorrect symbol of degrees Celsius in Fig. 4 (Line 484). We also moved the legend above the panel of figures in Fig. 4. We believe this will reduce potential confusion.

Figure 5. The symbol of micrograms is wrong. Which does p<0.0001 in the graph refer to? RBD alone?

We corrected the incorrect symbol for micrograms in Fig. 5.

The p-values given in Figs. 5, 6, and 7 of the submitted manuscript are the overall p-value for the ANOVA analysis (either One-way ANOVA for parametric data or Kruskal-Wallis for non-parametric data), and symbols (*, **, #, ##, etc.) represent the p-value of the post-hoc comparisons of group mean or group mean rank between the groups compared. The overall p-value was given to inform the reader that the initial ANOVA rejects the null-hypothesis (that differences in group mean/group mean rank are due to chance alone), which is a condition that must be met before any post-hoc comparisons are valid. To avoid confusion, we removed the overall p-value from all figures (Fig. 5, 6, 7), but the overall p-values are given within the text of the corresponding results sections.

Ref. 36 is incomplete.

We completed the reference (previously Ref. 36), which is now the following (Ref. 52, Lines 892-893):

  1. Stadlbauer et al., “SARS-CoV-2 Seroconversion in Humans: A Detailed Protocol for a Serological Assay, Antigen Production, and Test Setup,” Curr Protoc Microbiol, vol. 57, no. 1, p. e100, Jun. 2020, doi: 10.1002/cpmc.100.

Submission Date

04 June 2022

Date of this review

14 Jun 2022 19:16:23

Reviewer 3 Report

The manuscript entitled “Using dual toll-like receptor agonism to drive more Th1-biased response in a squalene and -tocopherol containing emulsion for a more effective SARS-CoV-2 vaccine” is very interesting and well designed the experimental apparoch for effective treatment of COVID-19. It needs some major edits before acceptable for publication.

The abbreviations used in the manuscript not uniform and not defined for some of the materials. Please make sure for uniformity.

In abstract, excellent formulation characteristics – change to desired formulation characteristics.

Use droplet size, these are emulsion – oil based dispersed preparations, so not particle size.

High homogeneity – is it PDI value high or stable PDI in observed conditions?

Material section missing in the manuscript.

Write the yield and confirmation of INI-2002 and INI-4001 by earlier references in the results section as well.

Write each tested parameter as separate section.

Write the IACUC approved protocol number.

Author Response

Reviewer 3

The abbreviations used in the manuscript not uniform and not defined for some of the materials. Please make sure for uniformity.

We have checked the abbreviations for concordance throughout the manuscript. Additionally, we have written out lesser known abbreviations in full, but commonly used buffers and reagents (DPBS, PBS, FBS, etc.) and techniques (cryo-EM, RP-HPLC, ELISA, etc.) remain abbreviated.

In abstract, excellent formulation characteristics – change to desired formulation characteristics.

We have changed the verbiage in the abstract to reflect the suggestion.

Use droplet size, these are emulsion – oil based dispersed preparations, so not particle size.

We changed the term “particle size” to “droplet size” throughout the manuscript when referring to emulsions.

High homogeneity – is it PDI value high or stable PDI in observed conditions?

The PDI is the stable PDI for the observed conditions. We added a note within the section 2.3 Physical characterization of emulsions to specify that the measurement is a stable measurement (Line 167).

Material section missing in the manuscript.

This is a reasonable suggestion from the reviewer. We added a materials section, “Materials 2.1,” within the larger section (Lines 107-113). In subsequent sections, we added vendor locations for assay materials and instruments to match standards of articles previously published in Pharmaceutics.

Write the yield and confirmation of INI-2002 and INI-4001 by earlier references in the results section as well.

This is an astute observation by the reviewer. We chose to clearly represent the agonist recovery as pre- and post-filtration rather than presenting the total yield or overall concentration through the entirety of the formulation process. Low formulation volume of the emulsions (1 mL) was a necessity due to the amount of INI-2002 and/or INI-4001 available at the time of the study, and consequently, we dealt with a dilution of 3-4x when processing with our high-pressure homogenizer (LV1). Briefly, the LV1 has input and output dead volumes of buffer (~2-3 mL in total), and when processing small volumes, this results in artificial dilution that would be avoided if allowed to prepare larger emulsion volumes or higher concentrations. Squalene-based emulsions have been made at manufacturing scale at 2x to 30x of the concentration without changing droplet properties, and we believe this would be a more appropriate use of total yield than our study.

We agree with the reviewer that framing our TLR4, TLR7/8 agonist recovery within the larger body of research will strengthen this manuscript. As such, we have added writing comparing TLR4, TLR7/8 agonist recovery with comparable dually encapsulated liposomes, as well as emulsions containing TLR agonists (Lines 425-460).

Write each tested parameter as separate section.

We feel that addressing the reviewer’s comment would make the “2. Materials and Methods” and “3. Results and Discussion” sections less cumbersome. We separated methods for each technique into sections. Additionally, we separated each figure or table into its own section. We think this reduces clutter to give the more clarity to the reader.

Write the IACUC approved protocol number.

We added the IACUC protocol number (015-19JEDBS ) to the manuscript body (Line 202).

Submission Date 04 June 2022

Date of this review 15 Jun 2022 21:08:34

Round 2

Reviewer 3 Report

No further comments.